# A Review of Single-Cell Microrobots: Classification, Driving Methods and Applications

**DOI:** 10.3390/mi14091710

**Published:** 2023-08-31

**Authors:** Yuhang Wang, Jun Chen, Guangfei Su, Jiaxi Mei, Junyang Li

**Affiliations:** School of Electronic Engineering, Ocean University of China, Qingdao 266000, China; yuhangwang0823@163.com (Y.W.); junchen082023@163.com (J.C.); guangfeisu@163.com (G.S.); laolang031@163.com (J.M.)

**Keywords:** single-cell microrobots, external drive, precise delivery, minimally invasive therapy

## Abstract

Single-cell microrobots are new microartificial devices that use a combination of single cells and artificial devices, with the advantages of small size, easy degradation and ease of manufacture. With externally driven strategies such as light fields, sound fields and magnetic fields, microrobots are able to carry out precise micromanipulations and movements in complex microenvironments. Therefore, single-cell microrobots have received more and more attention and have been greatly developed in recent years. In this paper, we review the main classifications, control methods and recent advances in the field of single-cell microrobot applications. First, different types of robots, such as cell-based microrobots, bacteria-based microrobots, algae-based microrobots, etc., and their design strategies and fabrication processes are discussed separately. Next, three types of external field-driven technologies, optical, acoustic and magnetic, are presented and operations realized in vivo and in vitro by applying these three technologies are described. Subsequently, the results achieved by these robots in the fields of precise delivery, minimally invasive therapy are analyzed. Finally, a short summary is given and current challenges and future work on microbial-based robotics are discussed.

## 1. Introduction

In recent years, with the development of theory and technology, micro-nano robotic systems have shown great flexibility, adaptability and safety at the micro-nano scale by virtue of their tiny size. Especially in the medical field, where micro-nano robots are expected to enter the brain, eyes, joints and many other areas that cannot be accessed by ordinary means for diagnosis and treatment. In order to work effectively in the microenvironment, the manufacturing process of micro-nano robots needs to consider several factors, such as controllability, visibility, functionality and biocompatibility, which are mainly reflected in the design and drive control [1,2,3,4,5].

Starting from the design, common micro-nano robots are mainly prepared by processing micro-nano materials. And the common methods for preparing micro-nano robots are physical vapor deposition technology [6,7], electrodeposition technology [8,9], controlled assembly technology [10], 3D printing technology [11,12] and biohybrid technology [13,14,15]. The more novel microbial-based micro-nano robots usually use bacteria [16,17], cells [1,18,19] or algae [20,21] as the robot itself or propulsion, which are highly biocompatible and biodegradable and can be degraded to noncytotoxic solutes in the biological environment [22,23]. Moreover, single-cell organisms usually have uniform spherical, spiral and linear structures, as well as sizes at the microscale, and they can be easily combined with other artificial materials. Using the properties of the organisms themselves, different artificial devices can be combined with single-celled organism to achieve different functions, such as targeted drug delivery [24], tumor inhibition [25], tissue repair [26] and so on. Therefore, microrobots composed of biological and artificial devices are simpler and cheaper to prepare than microrobots without biological components, work more stably and are safer, as well as show greater potential for future development [27,28,29,30].

In order to make micro-nano robots highly controllable in blood or other body fluid environments, two methods are usually used: self-propulsion [31,32] and external-field propulsion [1,33,34]. In self-propulsion, a chemical fuel is applied to the surface of the robot which chemically reacts with the liquid environment to create a large number of bubbles to achieve propulsion. In external field propulsion, the use of optical, magnetic, acoustic or electric fields enables freer and more permanent control of the robot. External field drives do not require chemical fuels in the environment, making them more suitable for biological applications than chemically driven robots. Moreover, external field drives have a high spatial resolution, and recent research on driving in three dimensions has broken through the limitations of one and two dimensions, giving micro- and nanorobots greater flexibility and functionality [35,36]. Microbotic systems can carry drugs and stem cells, targeting drug delivery to improve drug utilization after reaching a designated area or using the differentiation ability of stem cells to repair damaged sites. In addition, it facilitates easy capture, rotation and stretching of cells utilizing the microrobotic system, which facilitates the observation and study of cells and further advances the development of medicine, cytology and pharmacology in later clinical settings [37,38,39,40,41].

In this paper, we discuss and summarize recent research advances in micro- and nanorobotics by focusing on biological and design processing often employed in biotype microrobots, principles and effects of external field actuation and applications. Lastly, we discuss challenges and future research directions for biotype micro/nanorobots.

## 2. Types of Single-Cell Robots

Nature provides a rich database for science. Through natural selection, real-life organisms have evolved complex biomaterials with not only different structures but also a wealth of functions, such as hydrophobicity/hydrophilicity, reversible adhesion, biodegradability, regenerability, tropism, bioimpedance, fracture resistance, light weight, anti-reflectivity and autofluorescence (Table 1). Such biologically rich natural functions offer interesting options for robotic engineering. In the last decade, biohybrid microrobots, using microorganisms in combination with artificial structures, have received increasing attention.

### 2.1. Algae

Microalgae are eukaryotes that are easy to culture and they offer attractive properties for biomedical applications, including efficient propulsion (>100 μm/s), autofluorescence and phototropic guidance capabilities. They are minimally invasive and can move in inaccessible or unprecedented areas within the body, and they can serve as wireless transporters and drug delivery devices for specific localized areas [42,43].

*Chlamydomonas reinhardtii* are unicellular green microalgae and contain flagella. Flagellar propulsion is considered an efficient and fast propulsion mechanism in low-Reynolds-number hydrodynamic systems, such as small diameter vessels, small arteries, capillary sphincters, small veins and capillaries [44]. The two flagella of *Chlamydomonas reintiflora* move by bending to create plane waves. This microorganism is phototropic and also has a natural autofluorescence that allows label-free fluorescence imaging. They are biocompatible with healthy mammalian cells, are not cytotoxic, can move in physiologically relevant media and allow surface modifications to carry cargo on the cell wall. In 2005, Weibel et al. demonstrated the use of the single-cell photosynthetic microalga *Chlamydomonas reinhardtii* to transport micron-sized synthetic cargo in a light-controlled manner by UV irradiation with photolysis peptides attached to the microalgae to obtain controlled release [45]. A decade later, the phototropic motor control and steering capabilities of the multicellular green algae *Cryptobacterium hidradenum* and *Chlamydomonas reinhardtii* were proposed as an option that could be integrated into the design of biomixed microswimmers for biomedical applications [46]. Controlled movement of algal cells along arbitrary trajectories, including saw-wise and triangular movements, was achieved using the inherent phototropism mechanism of algal cells. In recent years, different morphological algal species, including spiral, ellipsoidal and spherical, have been utilized as biotemplates to fabricate magnetic microswimmers [47]. The use of micromotors in drug delivery is a promising and effective approach. Fangyu et al. combined the efficient movement and long life of natural algae and the protective ability of oral capsules to produce a good micromotor platform. The platform is able to improve cargo tissue motility in the gastrointestinal tract, can effectively self-propel in artificial intestinal fluid (SIF) and maintain sustained rapid movement over long periods of time [48]. Sitti et al. reported a biohybrid algal microswimming system (Figure 1A) that assembles a thin, soft, homogeneous coating around a motile microorganism by molecularly assembling the nonliving components around the cell wall. High fabrication yields are achieved through electrostatic interactions between the positively charged coating and the negatively charged membrane, laying the foundation for the development of a new generation of microalgal cargo platforms [49].

**Table 1 micromachines-14-01710-t001:** Microbial Robot Types and Functions.

Category	Advantages	Disadvantages	Classification	Features	Target	Application	Efficiency	Reference
Algae	Self-propelled, self-fluorescent, phototropic	Limited stability and life cycle of algae in the vivo	Rhine Clothier	Phototropic, naturally autofluorescent, biocompatible	Intestinal tract	Cargo handling	110 μm/s	[48]
			*Chlorella*	Contains chlorophyll, which overcomes tumor hypoxia	Tumor tissue	Relieving tumor hypoxia	80 μm/s	[50]
			*Spirulina*	Helical structure for noninvasive fluorescence imaging	Gastrointestinal tract	Tumor suppression by drug delivery	85 μm/s	[25]
			Diatom	Micro- and nanoscale porosity and large internal volume with biocompatibility	Intestines Road	Drug Delivery	30 μm/s	[24]
Bacteria	Self-propelled, perception of physiological gradients	May be cleared by the body’s immune cells	*Vibrio* lysozyme	Multiple self-propulsion methods		Bacterially driven liposomes	60 μm/s	[51]
			*E. coli*	Hydrophobicity, wettability, biocoupling and covalent binding	T1 breast cancer cells	Targeted drug delivery	22.5 μm/s	[52]
Cells	Biocompatibility and high drug-carrying capacity	Small size, difficult to operate manually	Red blood cells	Surface immunosuppressive properties	vein endothelial cells	Targeted delivery of anticancer drug	14 ± 2 μm/s,	[53]
			Sperm cells	Stronger athletic ability	Reproductive System		50 μm/s	[54]
			Neutrophils	Chemotaxis	Inflammation, tumor site		0.165 μm/s	[55]

*Chlorella* is a single-celled microalgae that can produce O_2_ through photosynthesis [56]. Notably, *Chlorella* is able to reduce endotoxemia in digestive disorders [57] and enhance host defense against peritonitis without side effects [58]. In addition, it contains a large amount of chlorophyll that absorbs a broad spectrum, leading to photosynthesis over a range of wavelengths. This property can be used in photodynamic therapy to generate reactive oxygen species (ROS) at 650 nm irradiation [59]. In recent years, an innovative approach to overcome tumor hypoxia using the natural photosynthetic system of *Chlorella* has been proposed. This method successfully delivered microalgal microrobots into tumor tissues to produce O_2_ in situ under red-light-induced photosynthesis to increase tissue oxygenation and alleviate tumor hypoxia [50] (Figure 1C).

*Spirulina* is readily trapped due to its helical structure, thus prolonging the retention time of the drug in the body. In addition, the intrinsic fluorescent chlorophyll produced by *Spirulina* (SP) allows for noninvasive fluorescence imaging without chemical modification [44,60], making the versatile SP particularly suitable for therapeutic and diagnostic applications. Using natural cyanobacterial SP as a drug carrier, Zhou et al. effectively loaded and delivered curcumin, improving the bioavailability and retention time of the drug in the gastrointestinal tract, thereby enhancing therapeutic efficacy. The morphological characteristics of SP were also verified to contribute to the ideal distribution of this drug delivery system in the intestinal tissues [25] (Figure 1D).

The silica cell walls of diatoms are colorless, transparent and hard. The frustule consists of two overlapping valves which are attached to the bundle, similar to a petri dish [61]. They have high specific surface area, high mechanical stability, biocompatibility, customizable surface chemistry properties, thermal stability and chemical resistance [62]. Due to these unique properties, diatoms have been widely used for controlled drug delivery [63]; A new mesoporous biodegradable silicon nanoparticle (SiNP) drug carrier obtained from natural diatom silica mineral available from the mining industry is presented in [24] Figure 1B.

### 2.2. Bacteria

Among different types of microbots, bacteria drive biological hybrids that excel due to their effective flagellar propulsion capabilities, ability to navigate in hard-to-reach body tissues and ability to sense physiological and pathophysiological gradients. In addition, developments in bacterial genetic engineering that enable their use as microrobots have also enabled a variety of capabilities, including reduced cytotoxicity and expression of targeted surface molecules [64]. Bacteria are one of the major groups of microorganisms that can be intimately and dynamically involved in the development of human health and disease. Bacteria have been utilized as promising delivery systems with multiple biomedical purposes [65]. Studies have shown that bacterial flagella are an effective biopropellant [66], and bacteria are being used as a transport vehicle for nanoparticles, facilitating their penetration and the subsequent release of the drug inside the tumor [51] (Figure 2A). Mostaghaci et al. proposed a method for attaching bacteria-like molecules to certain types of epithelial cells (membrane-expressing mannose) based on the affinity between lectin molecules at the tip of bacterial type I fimbriae and mannose molecules on epithelial cells (Figure 2B). This method is used for bioadhesion to epithelial cells and for targeted drug delivery to epithelial cells in the urinary tract or gastrointestinal tract [67]. Metin Sitti and his team found that viscoelastic properties regulating bacterial surface interactions may be the most critical factor for bacterial attachment and microswimmer motility. It was also shown that biohybrid microswimmers exhibit biased and directed motility under chemical elicitor gradients and magnetic guidance, respectively, with potential targeted drug delivery capabilities. Metin Sitti and his team also presented an optimized design and fabrication method for a high-performance, multifunctional bacterial-driven microswimmer [52] (Figure 2C). A multifunctional biohybrid microswimmer was also prepared by attaching erythrocytes loaded with anticancer Adriamycin drug molecules and superparamagnetic iron oxide nanoparticles (SPIONs) to the bioengineered motile bacterium *E. coli* MG1655 via a biotin–affinity–biotin binding complex [68].

### 2.3. Cells

With the advent of small robotic technologies, some exciting new applications such as targeted drug delivery and single-cell manipulation are beginning to gain traction. However, there are still some challenges to overcome before such technologies can be applied in medicine. Among them, propulsion and biocompatibility are the main challenges, as in the case of propulsion at microscopic scales where the Reynolds number is very low, making it difficult compared with some micro- and nanoartificial motors based on complex surface biofunctionalization and coating of synthetic materials, such as some particles based on self-thermophoresis propulsion using coated colloids [69] or biomimetic modifications [70]. Although it can increase their specificity, the controlled motility for driving tasks [71,72] have many incomparable advantages, such as the ability to encapsulate more drugs within their membranes, interact with other cells and tissues, combine the advantages of previous drug carriers (e.g., drug protection and selectivity) and the ability to penetrate some human tissues and provide specific driving mechanisms to ensure that it triggers drug release [73,74] at the right time and space. And this cellular carrier is more biocompatible than the micro- and nanocarriers mentioned above.

Cell-based micro-nanorobotic systems (e.g., erythrocytes, sperm cells, leukocytes and several other cells) are considered effective and biocompatible [70] for targeted drug delivery (Figure 3). The excellent escape mechanisms and biocompatibility of cells and microorganisms can be transferred to micro-nanorobots, thus contributing to targeted drug delivery [75]. Utilization of their membrane systems can protect payload from phagocytosis and rapid clearance by renal filtration, reducing drug toxicity in healthy tissues.

Among these cell-based carriers, red blood cells are of particular interest due to their wide availability, unique mechanical properties, surface immunosuppressive properties and versatile drug-carrying capacity. Since the 1920s, red blood cells (RBCs) have been extensively studied and employed as carriers, exploiting their immunosuppressive properties and multifunctional drug-carrying capacity [53], making possible the encapsulation of drugs within RBCs [76]. Combined with external magnetic alignment and guidance, red blood cell (RBC) motors incorporating iron oxide nanoparticles [77] maintain the stability and function of the original cells. And they show effective guidance and prolonged propulsion in various biological fluids with extensive antigenicity, transport and mechanical properties not possible with ordinary synthetic motors. In addition, quantum dots (QDs), the anticancer drug Adriamycin (DOX) and magnetic nanoparticles (MNPs) were coencapsulated into RBC micromotors to form a natural multi-cargo-loaded micromotor system for RBCs [53] with the ability to load and transport diagnostic imaging agents and therapeutic drugs, opening the door to the development of therapeutic diagnostic micromotors that can simultaneously treat and monitor diseases.

Similarly, to overcome the above-mentioned problems of synthetic material based micro- and nanoartificial motors, nature has simultaneously developed flagella, which have evolved for millions of years to work as micromotors. Among the microscopic cells that exhibit this mode of propulsion, sperm cells are considered to be fast and efficient. The current state-of-the-art in sperm-based microrobotics is a new type of microrobot made by coupling sperm cells to a mechanical load. Sperm robots use the flagellar motion of sperm cells for propulsion and therefore do not require any toxic fuel in their environment. They are also naturally biocompatible and show considerable motility, thus providing us with the option to overcome both propulsion and biocompatibility challenges. Because sperm can swim into tumor spheroids and fuse with somatic cells, sperm-based cellular micromotors are attractive for local delivery of highly concentrated drugs and for targeting cancerous lesions in female genital organs by noninvasive means. On this basis, Xu et al. proposed a sperm hybrid micromotor-based drug delivery system [54], with an anticancer drug (adriamycin hydrochloride) loaded onto motile sperm cells for effective transfer to the target cell/tumor, as well as a new drug delivery system [78], successfully loaded different anticancer compounds onto human sperm for effective treatment of 3D cervical cancer and patient representative ovarian cancer cell cultures. In addition, using the combination of sperm as a biological power source and miniature devices, a biohybrid microrobot [79,80] was developed by capturing bovine sperm cells within a magnetic microtubule, which provides a good source of inspiration for the design of sperm-templated artificial motors, like the self-assembly based bidirectional propulsion artificial flexible sperm-like nanorobot [81], which provides an important exploration of reliable nanorobot models at low Reynolds numbers.

In addition to the aforementioned microcellular robots that use red blood cells and sperm cells as carriers, neutrophils (a type of white blood cell) are also valid candidates for transporting drugs. As immune cells, they can autonomously target pathogens and eliminate them through phagocytosis [82]. Shao et al. developed a self-guided biohybrid micromotor using natural neutrophils [55] that inherits the intrinsic chemotactic capacity of neutrophils and moves autonomously along a chemoattractant gradient for targeted drug transport. There are also differentiated cells like those derived from monocytes (part of the innate immune system): macrophages [83,84,85]. Munerati et al. reported for the first time the drug encapsulation capacity of macrophages [86], using their most typical characteristics of phagocytosis of microorganisms and other particles for the transport of antiretroviral drugs to active viral replication sites, which can play an important role in the treatment of cancer and HIV-associated neurocognitive disorders. In addition, Lee et al. developed a hybrid immune cell-based microrobot [87] using a click-response-assisted immune cell targeting strategy to deliver drug-laden nanoparticles deep into the avascular regions of tumors, yielding promising in vivo results, which are attractive for increasing penetration and accumulation into tumors/tissues.

However, the cell carrier-based microrobot using its cell membrane coating only contributes to the recognition of the environment and thus movement, and cannot fully realize directional motion. Meanwhile using single-cell organisms such as algae and bacteria as the carrier of microrobots has its own physiological characteristics and unique movements which are prone to disorderly movement and are difficult to move on specific trajectories. Therefore, external acoustic and magnetic field stimulation drives are still required, which can provide very precise positioning and control for the microrobot while also compensating for the lack of its own power. Meanwhile, the use of external field-driven microrobots can be adapted and adjusted according to different types of external fields and working environments, with superb adaptability and flexibility.

**Figure 3 micromachines-14-01710-f003:**
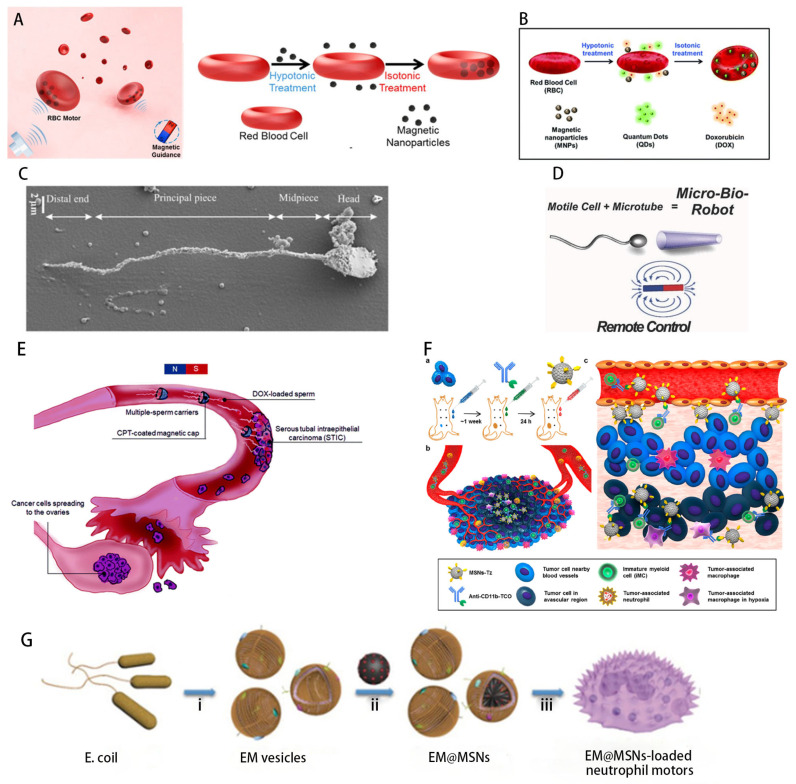
Cell-carrier-based microbial motor delivery system: (**A**) Preparation of natural red blood cells (RBCs) into functional micromotors with the help of ultrasonic propulsion and magnetic guidance [77]. (**B**) Preparation of a natural multi-cargo-loaded RBC micromotor system for therapeutic diagnostic applications, where quantum dots (QDs), anticancer drug Adriamycin (DOX) and magnetic nanoparticles (MNPs) are coencapsulated into RBC micromotors [53]. (**C**) A sperm-templated microrobot (IRONSperm) constructed by a unique electrostatic-based self-assembly fabrication method and described by a regularization-based Stokes–Leitz hydrodynamic model [79]. (**D**) A biohybrid microrobot developed by capturing bovine sperm cells inside a magnetic microtubule using motile cells as a driving force [80]. (**E**) A sperm-based drug delivery system targeting early plasmacytoid tubal intraepithelial carcinoma (STIC) lesions [78]. (**F**) Schematic diagram of a click-response-assisted immune cell targeting (CRAIT) strategy for enhanced tumor penetration of drug-loaded engineered nanoparticles (NPs) [87]. (**G**) The combination of live neutrophils with mesoporous *E. coli* membrane (EM)-coated silica nanoparticles (MSNs) developed by combining live neutrophils with mesoporous silicon dioxide nanoparticles (EMs). (i) Isolation and reconstruction of EM vesicles. (ii) Preparation of the EM-camouflaged MSNs (EM@MSNs) by a vesicle-fusion strategy. (iii) Incubation of EM@MSNs with neutrophils to prepare hybrid neutrophil micromotors [55].

## 3. External Field Drive

In recent years, great progress has been made in the study of microrobots, but one of the main challenges is that the provision and controllability of power for the movement of microrobots is important for them to perform further complex functions. In order to drive the motion of a microrobot and then be able to control it to perform specific functions in the human body, we need to provide it with the control information and the energy required to drive it. For microrobots of one millimeter or smaller size, it is difficult to load them with enough capacitor batteries [88] or connect an external power supply cord to provide drive power [89] like with traditional robots, and even if there is the possibility of this, it will largely limit the tiny robots’ movement space and speed. But, based on the small size, light weight, speed and high dynamics of the single-cell organism itself, it can stimulate itself for braking purposes by converting energy obtained remotely [90]. Many unicellular bodies and particulate materials can respond to physical and chemical stimuli that can be used to drive microrobots to achieve different forms of motion [91], such as grasping [92,93,94], rolling [95,96,97], crawling [98] and swimming [99,100]. Therefore, based on the current research status, the drive control methods for microrobots are physical control and other control methods, such as chemical control. Among them, physical control methods include optical drive, acoustic drive and magnetic field drive, while chemical control methods mainly include some self-propelled particles to generate endogenous power through some chemical or biological reactions such as diffusion electrophoresis, thermophoresis, etc., thus braking the microrobot (Table 2). It is important to note that the mechanism of motion of some particles in optically driven microrobots using photothermal catalysis has some crossover similarities with chemically driven ones.

### 3.1. Optical Drives

Unlike the open environment in the laboratory, microbots experience harsh conditions in the bloodstream, a viscous and fast-flowing environment that may be difficult to swim in. While microrobots like some of the traditional ones using chemically driven can move at high speeds, the chemical feedstock required is often toxic (described briefly in Section 3.4). Therefore, they cannot be used in blood vessels. In a nontoxic and noninvasive driving environment, optical methods have the advantage of remotely controlling cell motility without touching the cells, thus providing light-controlled or light-responsive microsystems [144]. With high instantaneous drive energy, light-driven microrobots tend to exhibit high-speed motion and are ideal for biomedical microrobot design. In addition, effective light drive requires only asymmetries in the structure or drive region of the microrobot, providing greater flexibility in the structural design of the robot [145].

In recent years, these micro/nanoparticles based on light-driven swimming have brought great hope and insight into the rapid development of next-generation biomedicine. Optical technology offers a promising approach for remote transport of microrobots to defined locations and controlled release. Mechanisms for micro/nanoscale particles driven by light are typically phototropism control, photothermal or photocatalytic propulsion, single-beam gradient force traps (optical tweezers).

#### 3.1.1. Phototropism

Many organisms are phototropic to light [131,132], i.e., move spontaneously and directionally toward or against light. Under this principle of stimulus-determined and environmentally driven directional movement, algae use their natural advantages of perception and drive to guide cell movement based on phototropic control using changing light signals [46,133]. And, using the chemical properties of algal cell walls to pick up microbeads and achieve guidance and transport [45] for algal cells to the long-distance transport and cooperative drive of micro-objects provides a new method. Inspired by natural cellular phototropism, the efficient self-propulsion of OsciBot [146], an all-soft swimming robot based on a hydrogel oscillator, mimics biophototaxis. Meanwhile, the Janus nanotree shown by Tang et al. [147] can be programmed to exhibit either positive or negative phototropism, similar to an optically tunable and optically programmable micro/nanomotors with electrically polarizable modes, which can transiently change the alignment direction and speed of semiconductor nanowires in a simple visible-light-exposed external electric field. This controlled light-directed motion is superior to state-of-the-art soft swimming robots.

#### 3.1.2. Photothermal or Photocatalytic Propulsion

With the invention of photoactive liquid crystal elastomers and materials, soft robots can be constructed using gels and elastomers that match the elastic and rheological properties of biological tissues and organs [148,149,150]. Among them, the photo-triggered thermophoresis of gels uses light-induced temperature gradients as a mechanism for the self-propulsion of micro/nanoscale particles, which can adjust their shape and motion strategy to environmental conditions under light stimulation and perform a variety of motion behaviors on demand, opening a new door for fuel-free manipulation of micro/nanoparticles. In contrast to a bubble microrobot controlled by optically induced heating [151] that can reach a speed of 4 mm/s in a salt solution, a tubular microrobot using 3D printing proposed by Yajing Shen’s team [145] and Wu et al. demonstrated a polymer multilayer microrobot driven by near-infrared (NIR) light [134]. Based on a photothermal mechanism, thermophoretic forces from an asymmetric temperature gradient in NIR light directly propel the microrobot. The speed advantage of the microrocket without chemical fuel is much greater than that of the above-mentioned bubble microrobot. Meanwhile, a recent one was based on light-induced temperature gradient to propel Janus particles [69,135]. As shown in Figure 4C, laser irradiation leads to a temperature difference between the hemispheres of gold–polystyrene Janus particles, resulting in an inhomogeneous distribution of polystyrene fluorescent particles around them [69]. In contrast, Yang et al. combined radiation pressure with photophoresis pull to push Au–polystyrene Janus particles as needed [135]. As a result, the temperature of the gold hemisphere of Janus microswimmers heated by directional light irradiation was about 7 K higher than that of the polystyrene hemisphere. This temperature difference can generate approximately 0.1 pN of photophoresis force and push Janus particles.

Meanwhile, the photocatalytic decomposition of water can generate electrolyte gradients and power particles, or the light energy can induce catalytic effects in the redox reactions inside the micro/nanorobots, which in turn generate chemical gradients or bubbles to further propel the nanorobots [154]. The BiOI-based Janus micromotor [155] can be propelled by photocatalytic reactions in pure water under environmentally friendly visible light without the addition of any other chemical fuel, and its propulsion process can be remotely controlled by the regulation of visible light power. A glucose-fueled Cu_2_O@N-doped carbon nanotube (Cu_2_O@N-CNT) micromotor designed by Wang et al. [152] can be activated by visible-light photocatalysis, which is nontoxic and biocompatible for excellent three-dimensional motion control in biological environments. Sen et al. [156] used titanium dioxide (TiO_2_) to convert light energy into mechanical energy by photocatalytic decomposition of organic matter under UV illumination mechanical energy, resulting in swarming behavior of surrounding non-photoresponsive SiO_2_ particles [157]. A TiO_2_–Au Janus photocatalytic micromotor fabricated by coating TiO_2_ microparticles with a thin film of gold by Dong et al. [153] is presented in Figure 4F. Only very low UV light intensity irradiation is required to cause propulsion by self-electrophoresis on the TiO_2_ surface, and in addition, photocatalytic propulsion can be turned on and off by incident light modulation.

#### 3.1.3. Optical Tweezer-Based Particle Manipulation

In the use of microrobots for precise drug delivery in the human body, cell surgery and some traditional biological applications, the cells must be oriented correctly. Therefore, the orientation control of biological cells is extremely critical. Optical tweezers (OT) represent a versatile tool for microrobotic micromanipulation [158,159,160,161,162]. In order to avoid damage to living cells by shining laser light directly on them [163], microrobots controlled by OT can be used for manipulation of cells or organisms on a microscopic scale; thus, optical tweezers provide precise access to the microscopic world of bioscience. Optical tweezers have also proved useful in physics [161,164,165,166], nanotechnology [167], chemistry [168,169], soft matter [170,171], spectroscopy [170] and nanothermodynamics [172,173], among many other fields. The basic optical tweezer setup consists of three parts (Figure 5A), trapping, imaging and position detection, and the basic principles of its design are shown in [159]. Microscopic particles (whose refractive index is higher than that of the medium in which they are embedded) can be trapped near the focal point due to the presence of scattering and gradient light forces. The scattering force is due to the radiation pressure of the beam and acts in the direction of beam propagation, and the gradient force pulls the particles toward the high-intensity focal point. Thus, using optical tweezers, a very wide range of particles can be trapped [174], from individual atoms to whole mammalian cells (Figure 5B).

However, direct capture of nanoscale targets using conventional optical tweezers is still very difficult and also insufficient to provide detailed conformational and chemical change analysis of complex biomolecular systems. In addition, they are coupled with some experimental limitations, for example, cell manipulation by laser is usually accompanied by photodamage. This leads to limitations in the further expansion of the already widespread OT applications. So, based on overcoming the limitations of conventional OT, four types of conventional optical tweezers have been specifically designed to address the above limitations, namely plasma optical tweezers (POT) [175,176], photonic crystal optical tweezers (PhC OT) [177], femtosecond optical tweezers (Fs OT) [178,179] and optical tweezers incorporating various fluorescence techniques [180,181,182]. Among them, manipulation of nanoobjects is achieved by plasmonic optical tweezers (POT), which use local surface plasmon to generate optical traps with enhanced capture potential, and photonic crystal optical tweezers (PhC OT), which achieve the same goal by employing different photonic crystal geometries. Femtosecond optical tweezers (Fs OT) are constructed by replacing a continuous wave (cw) laser source with a femtosecond laser and are expected to significantly reduce damage to living samples.

Over time, optical tweezers have rapidly evolved into a widely used tool in many fields [161], including single-molecule biophysics, microchemistry, statistical physics and laser cooling and trapping of atoms, to name a few [183]. Today, OT is widely used to study cell biology [184] (Figure 6), for example, Paterson et al. used optical tweezers in combination with designed PDMS devices to successfully prepare and isolate single cells [185]. Optical-tweezers-based laser trapping are used for studying the ionization of living cells [186,187]. Optical tweezers are employed for nonlinear elastic and viscoelastic deformation of erythrocytes [188]. Optical tweezers are utilized to probe the dynamics of agglomerative interactions between erythrocytes and some mechanical characterizations [189,190,191,192,193]. Enhanced sorting and manipulation of single cells and some microparticles by optical tweezers in microfluidic devices is discussed in [136,137,138,139]. Hollow Annular Core Fiber (HACF)-based single-fiber tweezers were developed for rare cell manipulation and aseptic transport [140]. Medical diagnosis by optical tweezers is possible, for instance for measuring changes in the properties of red blood cells (RBCs) [141,142,143].

Nevertheless, due to their complex and expensive setup, until now, it has been difficult to achieve out-of-plane operation of microrobots using planar OT.

### 3.2. Acoustic Manipulation

Since their introduction, optical tweezers have shown excellent capabilities but also insurmountable disadvantages. One is that lasers can heat the manipulated object and even change the activity of cells [195], bacteria and other biological objects. The other is that lasers are limited by penetration and cannot be operated in vivo; therefore, the use of optical tweezers in the biological field is limited. To compensate for the shortcomings of optical tweezers, new control technologies such as acoustic tweezers and magnetic tweezers have been born. Acoustic tweezers have the widest control range (10 nm–100 μm) with the lowest power ($10^{-2}–10\;W/{cm}^{2}$) and also possesses selectivity [196]: the ability to select a single object in a group for operation. In addition to its excellent performance, ultrasound is also widely used in medical applications, such as ultrasonography, imaging, ultrasonic lithotripsy, etc. Acoustic tweezers use the same wavelength of sound waves as the medical use of ultrasound, so they have a high level of safety. Especially, when compared with magnetic tweezers, acoustic tweezers do not require the introduction of auxiliary particles (ferric tetroxide), which can alleviate concerns and enable early in vivo manipulation.

There are three types of acoustic tweezers according to principle, standing-wave tweezers, traveling-wave tweezers and acoustic flow tweezers. The process of acoustic flow tweezers works by a steady flow generated by the absorption of acoustic energy by a liquid, thus indirectly manipulating particles in solution. This flow, called acoustic flow, is most commonly generated by oscillating microbubbles or oscillating solid structures [128,129,130]. Although acoustic flow tweezers are simple to operate, they are less maneuverable, and usually acoustic flow tweezers use a piezoelectric ceramic placed on the side of the microfluid, which can only drive particles in one direction and cannot work in the complex human vasculature. Therefore, it is expected that the control of cells, bacteria and other single-cell organisms in vivo drug delivery and targeted therapy is achieved by standing-wave tweezers and traveling-wave tweezers, and the principles of each are described below, along with research progress.

#### 3.2.1. Traveling-Wave Tweezers

Traveling-wave tweezers are typically used to trap particles and control their movement by varying the phase difference in an array of transducers to create an acoustic potential trap in space. Transducer arrays with phase differences have different phases in the same plane, and when superimposed, the center has almost zero amplitude, and the highest energy point is the surrounding annulus, so that particles can be trapped in the center. Traveling-wave tweezers require a minimum of four transducers to control particle movement, sorting and rotation [119]. The greatest advantage of traveling-wave tweezers is the ability to separate individual particles in a population for three-dimensional manipulation, which is commonly used for cell sorting, observation and cell printing, tissue engineering, etc. [120].

Array-based traveling-wave tweezers can generate complex sound fields, allowing for diverse operations, such as planar movement [119], rotation [121] and levitation [122]. Yang et al. [123] built a 3 MHz, 64-element (8×8) 2D planar ultrasound arrays, using a computer to calculate the phase and amplitude of each element. And by exciting the arrays with the calculated signals, focused vortex and double-trap fields can be generated. These acoustic fields are used in conjunction to achieve free movement and levitation, and the maximum velocity during the planar movement can reach 564 µm/s. Macrophages are immune cells that are capable of engulfing broken cells and pathogens. Macrophages are ideal drug carriers because of their phagocytic ability [124]. Hiep Xuan Cao [125] use a phase-modulation method to form an acoustic potential well at the focal point, and macrophages were manipulated to follow an arbitrary trajectory in a spherical three-dimensional space with a diameter of 4 mm. The macrophages were manipulated to move along an arbitrary trajectory in a spherical three-dimensional space of diameter. In addition to phagocytosis, macrophages also have the ability to target tumors and can later be processed to carry drugs to target tumors.

Usually, traveling-wave tweezers need to adjust the phase difference in the transducer array to move the position of the potential well, which requires strict array global cooperation and usually requires the help of computer algorithms to achieve, which is more difficult to implement. Using a mechanical stage in the X–Y plane is a simpler approach, but this requires a smaller sensor array size, so achieving integration is a new direction. Baudoin et al. [126] deposited metal electrodes on the surface of a piezoelectric substrate, which triggered local vibrations of the piezoelectric substrate under the excitation of a signal, thus generating acoustic eddy currents. The integrated transducer has a diameter of 21 mm and an output frequency of 4.4 MHz (Figure 7A). In their later work [127], the size was further reduced. Together with an adjustable mechanical platform, the tweezers are able to move a radius of 75±2 μm polystyrene particles to a defined position. Moreover, the acoustic tweezers instantaneously capture human breast cancer cells while giving a drive signal (7+1 μm) and moving in a predefined trajectory. No disruption of cell activity was observed in subsequent cell observations.

#### 3.2.2. Standing-Wave Tweezers

The principle of standing-wave tweezers is to use the acoustic radiation force in the standing-wave field to push the particles to the node or antinode—the energy extremum point—and adjusting the frequency can change the acoustic field distribution, thus driving the particles to move. Depending on the way of forming standing waves, standing-wave tweezers can be divided into BAW and SAW [110]. BAW generally uses one or more group of IDTs (fork-finger transducers) to emit sound waves superimposed in space to form standing waves. The IDTs produce acoustic surface waves that propagate along the surface, so they are mostly used to generate one- and two-dimensional standing-wave fields. IDTs have a higher frequency and can cover 10 MHz~5 GHz. So, IDTs are an excellent choice for particle transport and separation, bioengineering and cellular and microbial manipulation [111,112,113].

Jooss et al. [113] used standing-wave acoustic tweezers to manipulate particles in zebrafish embryos and demonstrated that acoustic radiation forces can keep particles stable in flowing blood. Similarly, in Maramizonouz et al.’s [114] experiments, the captured particles were not only not dispersed by the blood but were more closely aligned along the blood flow. In contrast, the experiment by Leshno et al. [115] applied standing-wave acoustic tweezers to an eye and succeeded in collecting and discharging the particles dispersed in the eye. These studies suggest that standing-wave acoustic tweezers should have great potential for real in vivo manipulation. The selectivity of acoustic tweezers is wide and can be adjusted according to the size of the manipulated object and therefore is highly adaptable. Ding et al. [111] placed two pairs of orthogonal IDTs on both sides of the PDMS channel. The acoustic radiation force drives the particles to nodes or wave bellies in the acoustic pressure field, where cells, nematodes or particles trapped in the pressure nodes can be manipulated freely in both dimensions. Moreover, the power of the acoustic tweezers is very low, and in this system, for about 30 μm/s particle velocity, manipulating 10 μm polystyrene beads requires a power density of about $50\;nW/\mu m^{2}$. This is lower than optical tweezers by $1\times 10^{7}$ times and $1\times 10^{2}$ times less than magnetic tweezers [116,117]. Therefore, acoustic tweezers are easier to miniaturize and more biocompatible than other control methods.

As mentioned above, although SAW has demonstrated accurate control and spatial resolution in one and two dimensions, it is still somewhat difficult to achieve operation in three dimensions. Guo et al. [118] proposed 3D acoustic tweezers: two pairs of IDTs embedded in a PDMS cavity substrate generate a 2D standing-wave field on the surface of the substrate upon activation. Meanwhile, the transducer vibration triggers a 3D acoustic flow in the cavity, generating a 3D capture node under the acoustic field flow field. Using this system, single suspended mouse fibroblasts were successfully captured and printed in patterns such as numbers and letters, and the cells also underwent division and value addition after a period of time; thus, the system is expected to enable cell building engineering at the microscale.

### 3.3. Magnetic Drives

One of the main challenges in developing bolus-free microbots is their propulsion and manipulation. In the absence of external factors, the motion of a given microorganism is usually random. Therefore, it is crucial for biohybrid microrobots that their motion is controlled and manipulable. Controlling the motion of biohybrid microrobots can be achieved in two ways: (i) by the microorganism’s use of surrounding chemical energy thrusts (e.g., aerotropism, ph-tropism, phototropism) and (ii) externally by manipulating artificial chambers connected to living microorganisms (e.g., magnetic field, acoustic wave control, or light field control) [49]. Magnetic and optical fields often enable remote, dexterous, precise, fast and robust actuation and control of microrobots. While optical actuation is suitable for environments with much lower surface penetration depth requirements or transparency, magnetic actuation has unique potential for medical applications within nontransparent tissues with high penetration depths [197]. Magnetic drive methods can produce higher velocity motion and force output than optical drives depending on the design and drive parameters, and it is easier to generate high degrees of freedom using magnetic drives than optical drives due to the ease of generating magnetic fields and gradients in three-dimensional space. In addition, acoustic waves dissipate easily, and the effect dissipates faster after driving the microrobot; so, magnetic drives have significant advantages in terms of the degree of precision and powerful control compared with acoustic drive control. Because magnetic fields can continuously transmit power to robotic devices in a wireless manner with high penetration capability through the human body harmlessly, even at relatively high magnetic field strengths, magnetic actuation has attracted special attention. Magnetic micro-nanorobots can be remotely controlled so that they can use magnetic fields for high-precision propulsion in complex biofluids. Their potential for controlled navigation in cavities that are difficult to reach by the human body makes them promising microrobotic tools to diagnose and treat diseases in a minimally invasive manner.

In magnetic microrobots, micromachine bodies can be made of either hard magnetic materials or soft magnetic materials. Magnetic materials are categorized as ferromagnetic (and subferromagnetic) materials [198], paramagnetic materials [199] and antimagnetic materials [200,201]. Single-cell organisms are not inherently magnetic, and in recent years, a variety of magnetic materials have been used to magnetize single-cell micro- and nanorobots, such as ferric oxide nanoparticles [47], magnetic polystyrene (PS) particles [202] and compounds of terbium [203]. A biohybrid magnetite microrobot that is imagerable, biodegradable and has selective cytotoxicity against cancer cells was reported by Zhang et al. The fabrication employed a simple one-step approach using *Spirulina* as a biotemplate and a simple dip-coating method to bind tri-iron tetroxide nanoparticles (NPs) to the surface of *Spirulina*. The prepared biohybrid robot was able to perform robust navigation in various biological fluids for noninvasive tracking. It also exhibited desirable biodegradability, remote diagnostic sensing capability and anticancer potential [47]. Despite advances in the use of different species of microalgae as robot drivers or templates, the motion and function of algal microswimmers when composed and driven by live microalgae have not been studied in depth. Sitti et al. present the design and fabrication of a biohybrid algal microswimmer for biomedical applications, which consists of an intact microalgae, Cladosporium reinhardtii, attached to a polyelectrolyte (PE) functionalized 1 µm diameter magnetic polystyrene (PS) particles. Using a non-covalent electrostatic interaction mechanism, the magnetic PS particles were deposited on the final positively charged polyelectrolyte layer and incubated with negatively charged microalgae for 30 min to prepare biohybrid algal microswimmers [202]. Santomauro et al. presented a proof of concept to achieve the magnetotropic behavior of microalgae by incorporating terbium into the microalgae [203]. Magnetized microalgae are easily tracked in humans by luminescent imaging tools due to their intrinsic self-fluorescent properties and the photoluminescence of the incorporated Tb^3+^.

Magnetic drive methods all require auxiliary drive/control and imaging equipment to remotely drive and control the mobile microrobot. In the case of magnetism, an external magnetic drive system is required (the magnetic manipulation system consists of a set of permanent magnets or electromagnets [200,204] as the source of the magnetic field), robot positioning and tracking systems, control hardware and computers, user interfaces and imaging systems. Drive and control the closed-loop motion of a microrobot for a given application. In many applications, a swarm of microrobots needs to operate in parallel to deliver a large number of goods (e.g., in a targeted drug delivery application). Therefore, in these application scenarios, many microrobots need to be driven and controlled individually or integrated. Swarm control of magnetic microrobots has also become possible in recent years. The mean and variance of swarms can be controlled at solid surfaces [205], water-air interfaces [206] and fluids [104,207] using dynamic self-assembly methods and overall motion behavior controlled by external magnetic fields/gradients.

Magnetic drive systems can be classified as electromagnetic drive systems and permanent magnet systems [208]. Magnetic drive systems can include custom-designed or Helmholtz coils, magnetic resonance imaging (MRI) gradient coils within magnetic resonance imaging scanners, coil arrays, or permanent magnet systems [197] (e.g., single or two dipole permanent magnets, Halbach arrays).

#### 3.3.1. Permanent Magnet System

Permanent magnet systems have low heat generation, convenience, and strong magnetic fields. Many researchers have reported the use of permanent magnets for the control of magnetic micro/nanorobots. Sitti et al. proposes about 100 permanent magnet array-based remote-propulsion microrobotic systems that enable cylindrical magnetic microrobots to navigate soft tissues through continuous penetration [101] (Figure 8A). Maoney et al. proposed a three-degree-of-freedom (3DOF) closed-loop position and two-degree-of-freedom open-loop orientation control method using a rotating permanent magnet to drive a tetherless magnetic device for a gastric capsule endoscope [102]. Since a single permanent magnet does not provide sufficient magnetic field strength, a system of two or more rotating permanent magnets can be used. In 2017, a new magnetic control system was proposed. This system enabled the design of a system with eight permanent magnets that can create fields and field gradients in any direction, allowing remote and precise control of microrobots in three-dimensional space [103]. The strength of the magnetic field generated by the permanent magnets is constant, and the flexibility of the permanent magnet system is compromised by the degrees of freedom of the robot arm, in addition to the fact that the magnetic field source of the permanent magnets is always on.

#### 3.3.2. Electromagnetic Drive System

The electromagnetic manipulation systems utilize an array of electromagnetic coils to generate a magnetic field. Adjusting the current in each coil enables these systems to generate different magnetic fields or change the flux and gradient. These coils can freely vary the strength and type of magnetic field by changing the amplitude and direction of the current, thus enabling precise control of micro/nanorobot motion. Helmholtz coil pairs can produce uniform magnetic fields, and three-dimensional Helmholtz coil systems can produce a uniform rotating magnetic field [107]. Salehizadeh et al. proposed two new strategies for controlling the motion of rotating magnets based on magnetorheological interaction [209]. Sun Dong and other teams proposed the core shape design of the electromagnetic coil and developed a drive system based on the magnetic gradient field, which can realize the operation of microrobots [210]. They also generated a gradient-based rotating magnetic field by sequential excitation of the individual coils of the solenoid coil system. Through this method, a high magnetic field gradient concentrated at a specific target site is generated, attracting microsubject groups to automatically converge to the target site from different directions [211]. Thus, a variety of magnetic fields suitable for different microrobotic motion control can be generated. Nelson et al. investigated controlling the in vivo swimming of a small bacteria-like robotic flagella swarm [104]. Li et al. used three orthogonal electromagnetic coils to generate a rotating magnetic field, and navigational motions in a bull’s eye sphere were achieved (Figure 8B) [105]. Cai et al. used a custom three-axis Helmholtz coil system to drive sperm-like nanorobots and tested their propulsion performance [81]. Jun Cai’s group also investigated the propulsion and corresponding photocatalytic performance of biohybrid magnetic microrobots using a three-axis Helmholtz coil system [106]. Other electromagnetic systems such as Maxwell coil systems and various types of coils [108,109] were developed to generate adaptive magnetic fields for different functions.

### 3.4. Other Stimulus Drivers

In addition to the mainstream physical-based drives, there are currently some other drive-controlled self-propelled particle designs [212,213,214] where the self-propelled particles move themselves by generating endogenous power [215] from the energy of the environment (e.g., chemical, electrical [216,217] and thermal [218,219]) through some chemical or biological reactions. Because of the continuous chemical energy to be obtained from the environment, chemical particles or catalysts must be attached, leading to the fact that the structure of such microrobots is always asymmetric. Designing self-propelled particles of a few millimeters or smaller is not straightforward, because as the size decreases, effects such as Brownian collisions, viscous drag and surface phenomena (diffusion electrophoresis, electrophoresis, thermophoresis, etc.) become dominant. And motors, for example, may not be as effective as their corresponding large analogues due to the limitations of the aforementioned microscale effects. Therefore, new technologies for powering (or fueling) these objects are needed to facilitate their propulsion. Examples include Janus particles that move along a localized temperature gradient, emulsion hydrogel motors (E-H motors) (Figure 9) that are efficiently propelled by generating bubbles through thermal stimulation, and electrostatically anisotropic hydrogel actuators [220], the first two of which are based on autothermophoretic actuation [69,221]. Some particles also move using diffusion electrophoresis induced by concentration gradient differences on the surface of particles [147] and the Belousov–Zabotinsky reaction [222]. A number of groups also use intact living cells as loads [80,223] or integrate biomolecular motors or enzymes with synthetic components to engineer hybrid self-propelled particles [224,225,226,227]; most of these particles exhibit continuous translational motion. Based on chemical reaction drives, objects such as synthetic robots from small molecules to millimeter size enhance motion by chemical drives, such as versatile methods based on efficient control of 3D microflow and partial dissolution of particles to generate complex-shaped three-dimensional multichamber (3D-MC) particles [228]. The conversion of chemical energy into a nanorobot drive based on redox reactions is a very popular scheme. In particular, the decomposition reaction of hydrogen peroxide to produce water and oxygen is the most commonly used method [229]. Under the principle of this decomposition reaction, it is widely used in nanorobots in the form of bimetal nanorods [230,231], spherical particles [232,233,234,235], combinatorial microtubes [236], etc., which are self-propelled by bubbles of hydrogen peroxide decomposition. However, the degree of chemomechanical coupling and diffusion enhancement is strongly size-dependent and disappears as the size of the swimmers approaches the molecular scale.

Therefore, in some special cases limited by the manipulated driving environment, the choice of the Marangoni effect as a driving force for microrobots is a better alternative to the use of chemical stimuli [237,238]. These gel-based, self-propelled particles driven by the Marangoni effect convert light into thermal energy, which in turn induces interfacial tension gradients, creating a periodic propulsion pattern that can be designed to move in complex, multidirectional, preprogrammed trajectories. For example, the proposed ethanol-driven microrobot with photonic colloidal crystal hydrogels [239] enables long-term propulsion of the microrobot. Depending on the ethanol release rate, the propulsion behavior can change from continuous motion to pulsating motion.

The advantage of such other driving methods like chemical energy is that the microrobot does not need to be controlled all the time and only needs to be guided to the final target position by some external fields. However, at the same time, microrobots based on these driving methods have many drawbacks, such as difficulty in controlling the direction of motion, the movement process is easily disturbed by other ionic media, leading to a great limitation in the continuity of motion and the microrobot at a later stage may lack power as these chemical reactions proceed. Furthermore, another major challenge for its application in living organisms is the biocompatibility and safety of the “fuel” and reaction products. These issues deserve further exploration in the future.

## 4. Use Analysis

### 4.1. Cell Analysis

To meet this need, researchers have developed several cell sorting methods over the past decade, such as magnetically activated cell sorting [240], optical tweezers [136], electrophoresis [241] and microfluidic systems [242,243,244].

Based on the inherent good phototropism of cells, the combination of submillimeter-sized cargoes with stimulus-responsive biomotors in autonomous microsystems has enabled the unidirectional transport of submillimeter-sized cargoes [46,144]. Meanwhile, optical tweezers (OT) have been widely used in manipulating individual biological cells and performing complex biophysical/biomechanical characterization [195] for cell separation and delivery of particles or cells using the sterile (noninfectious) properties of microchannels in microfluidic systems [140,245] and utilizing the noncontact (noninvasive) properties of optical tweezers to achieve precise operations, such as cell sorting and selective capture in microfluidic systems [137,138,139,185]. For example, Sun et al. [136] employed an integrated universal single-cell cell sorting tool based on optical tweezers, microfluidic chips and imaging processing technology that can be used for high-precision processing of small cell populations for sorting and achieving parallel sorting of multicells. Additionally, mechanical characterization and rheological characterization of single cells [188,193,246,247,248,249], quantifying cell-to-cell interactions, can help detect and diagnose different physiological and pathological conditions in the human body and have a significant impact on the timely and effective treatment of human diseases. For example, Michael C. DeSantis’ team [250] optically captured individual virus particles in a suspension and then delivered them to the host cells and combined them with fluorescence imaging to effectively measure manipulated viral attachment and dissociation of [111] the host cell surface. This opens new avenues for studying viral particle-cell interactions. In order to overcome the problem of the difficulty in trapping and manipulating individual cells in crowded environments such as blood vessels and lymph nodes, Gong et al. [251] introduced an optical shielding scheme formed by a far-field Bessel beam to isolate the target cells from interference and finally succeeded in trapping and manipulating individual blood cells with a double-well optical tweezer.

The above methods are often limited by other factors that can affect cell survival and proliferation capabilities, such as excessive volume and excessive power. In the acoustic field, cells of different sizes and volumes are subjected to different degrees of acoustic radiation forces, resulting in different degrees of lateral drift and eventually being pushed to different outlets. And acoustic waves have been widely proven not to affect cells; so, acoustic-based cell sorting techniques have very promising applications. Xiaoyun Ding’s [252] SSAW-based cell sorting device consists of a single-layer polydimethylsiloxane (PDMS) channel and a piezoelectric substrate with a pair of orthogonal IDTs. By tuning the frequency, the system can precisely move the PNs/ANs in the lateral direction perpendicular to the cell flow, thus driving the captured cells folate to the designated outlets. One-milliliter cells with an abundance of fewer than 1000 in the sample are considered rare cells [253]. Rare cells are important for clinical treatment, prenatal diagnosis and medical research. Wang et al. [254] used BAW to sort rare cells and was able to achieve a sorting efficiency of 80%, satisfying the cell content used for analysis.

Measurements of cellular mechanics are essential to better understand cellular responses during the progression of certain diseases and to identify the nature of the cells. Hwang et al. [255] used acoustic tweezers to precisely attach fibronectin-coated microbeads to human breast cancer cells and then stretched the cell membrane by remotely pulling the cell-attached microbeads with an acoustic trap and found that highly invasive breast cancer cells exhibited greater deformability than weakly invasive breast tumor cells. Recently, Lee et al. [256] used an artificial neural network (CNN) to measure the rate of cell area change at each pressure level after deforming the cells using acoustic tweezers and then applied a multilayer perceptron (MLP) to learn the correlation between the rate of cell area change according to the pressure level and the deformability of the cells. The system was trained to discriminate against invasive breast cancer cells and noninvasive breast tumor cells. For better single-cell analysis, precise rotational manipulation of individual cells or organisms is essential. Ahmed et al. [257] captured microbubbles in predefined sidewall microcavities within the microchannel, and, driven by the acoustic field, the captured microbubbles oscillated, generating a steady stream of microacoustics that were used to rotate colloids, cells and whole organisms (*Cryptobacterium hidradi*). This system provides an excellent tool for single-cell mechanics biological studies.

A strong magnetic field induced by an external permanent magnet or electromagnet was used to separate magnetic beads attached to target cells. Bis–Zn–DPA-modified magnetic nanoparticles (MNPs) were used by Lee et al. to remove MNPs bound to *E. coli* from bovine whole blood using a magnetic microfluidic device [258]. Martin et al. described a nondilutive high-gradient magnetic separation (HGMS) device for removal of malaria-infected erythrocytes [259]. Boyle et al. proposed a MACS method based on coil–coil peptide interactions, where isolated cells can be readily separated from iron oxide particles by trypsinization, and this MACS system can effectively facilitate cell sorting [260].

### 4.2. Organ and Tissue Analysis

In order to cure patients who need to replace organs or tissues, it becomes important to synthesize organs or tissues artificially. Controlling single cells precisely and without affecting their activity so that they can divide and proliferate in the subsequent time is the most critical part of the above goal [261].

Currently, most OT-based experiments can only be performed in vitro in the form of cells, however, due to the complexity of the in vivo environment, studies performed in vitro may not accurately reflect the biological activity in vivo. If optical principles are applied to perform some in vivo manipulations, some difficulties are difficult to be completely resolved at this stage. First, in vivo optical force calibration is more complex than in vitro due to the variability of the capture environment in different tissue locations. Second, biological tissues have strong absorption and scattering properties, which makes it difficult to deliver sufficient capture intensity deep into the tissue without causing thermal or optical damage. Therefore, the penetration depth of optical actuation is very limited, which is the main drawback of optical actuation methods in medical applications. Only light sources with larger wavelengths such as NIR light can penetrate up to 1–2 cm under the skin tissue, allowing for some more limited operations in vivo. For example, the capture and manipulation of cells in living animals in vivo has been achieved using infrared light tweezers [262,263,264].

The acoustic field is relatively mild, has no effect on cell activity and its spatial resolution is sufficient to control cells for manipulation, making it a good choice for cell observation, printing, tissue construction and organ synthesis using acoustic tweezers [265,266,267]. Garvin et al. [26] developed the ultrasonic standing-wave field (USWF) technique for tissue engineering and experimentally showed that acoustic radiation forces can organize mammalian cells and cell-associated proteins into discrete bands within collagen hydrogels and that the cell bands can be stable for more than 20 h. Using this technique, tissue building, cellular functionalization and extracellular matrix remodeling can be achieved in three-dimensional artificial constructs. Gesellchen et al. [268] used a heptagonal acoustic clamp device to acoustically trap cells to generate dynamic cell patterns, generating different geometries of Schwann cells by selectively activating paired piezoelectric transducers located on either side of the heptagonal device.

In the above methods, cells are guided to the desired location by applying forces to the cells, which can be optical, magnetic, electrodynamic or fluidic in addition to acoustic radiation forces. There is also a method to efficiently construct tissues or organs—3D bioprinting technology. Three-dimensional bioprinting deposits and patterns cells in three main ways: droplet-based bioprinting, microextrusion [269,270] and laser-assisted bioprinting [271,272,273,274]; the last way includes digital light processing (DLP) and laser-based printing. Of these, acoustic-radiation-based droplet printing enriches cells by dropping material from a cartridge onto a substrate in a continuous stream. Droplet printing is the most commonly used printing method due to its high material usage, high printing accuracy and simple design [275]. For example, Gong et al. [276] developed an acoustic droplet printer that uses nozzle-free open biocartridges programmed to automatically print cell-filled droplets onto a hydrophobic PDMS receiver layer. This platform enables rapid preparation of size-controlled organoids that retain similar activity to the parent tissue at two weeks.

### 4.3. Drug Delivery

The scale of micro-nano-robots capable of navigating to tissues and other body cavities that are currently inaccessible can provide small medical tools for biomedical applications, such as disease monitoring, targeted drug delivery and minimally invasive surgery [44,277,278,279,280]. As used for human therapeutic applications, micro-nanorobots are required to have a lack of cytotoxic response, biodegradability, chemical stability, noninvasive in vivo tracking, targeted drug release and stimulated response release. In recent years, micro-nanorobots have been used for the treatment of several human organs, such as breast, stomach, intestine, etc. The application in vivo is still in its initial stage. High drug delivery efficiency, prolonged drug release time and low systemic toxicity are effective weapons for drug delivery systems to defeat metastatic breast cancer. Zhou et al. developed a biohybrid strategy of DOX-loaded *Spirulina* to target lung tissue and effectively inhibit lung metastasis of breast cancer [281] (Figure 10E). The oral to the gastrointestinal (GI) tract has been one of the most widely used drug delivery methods due to its high patient compliance, noninvasiveness, simplicity and low cost. Despite the potential advantages of oral drug formulations, they face several obstacles in the GI tract, including poor stability of gastric juice, limited drug interaction with the intestinal lining and low solubility [282]. Wang et al. reported an algal-motor-loaded capsule system that combines the efficient and persistent motility of the natural alga *Rhizopus rheinlandia* in the small intestine with the protective ability of an orally administered capsule, resulting in extended retention time within the intestinal mucosa and thus greatly improving gastrointestinal delivery [48] (Figure 10F). The helical structure of *Spirulina platensis* is not only more easily captured by the intestinal villi but also more readily adheres to the intestinal wall, thus prolonging the retention time of the drug in the intestine. A microalgae-based oral drug delivery system (SP@curcumin) was constructed using the natural cyanobacterium SP as a drug carrier for effective loading and delivery of curcumin to improve the bioavailability and retention time of the drug in the gastrointestinal tract, thereby enhancing therapeutic efficacy [25] (Figure 10G). Local hypoxia of tumors in the human body is a serious obstacle to cancer treatment, leading to a significant reduction in efficacy. Improving oxygenation and overcoming hypoxia in hypoxic tumor areas should significantly improve the efficacy of radiotherapy (RT) and photodynamic therapy (PDT). Thus, reoxygenation of hypoxic tumors would be an effective way to overcome the resistance of conventional cancer therapy to hypoxia-based therapy [283,284]. Recent studies have proposed an innovative approach to overcome tumor hypoxia using a natural photosynthetic system based on *Chlorella* in situ to generate O_2_ [50]. Drug delivery efficiency is also a current challenge. In this regard, Peters et al. have recently made significant progress by developing robust torsional spiral microrobots that achieve a surface area increase of more than 150% without compromising the propulsion speed [285]. SSA, a metric to evaluate the surface performance of structures, has not been very satisfactory. The magnetite porous hollow spiral microrobots proposed by Zhang et al. overcome this deficiency by having an ultrahigh SSA with values up to 52.22 m^2^g^−1^. The double increase in SSA is attributed to its hollow core structure. In addition, the microscopic roughness of the shell and the porous nature contribute to a further increase in SSA [286].

There is a long history of using bacteria to treat diseases such as cancer [287]. In recent years, the therapeutic effects of different bacterial species on tumors, diabetes and enteritis have been studied. Bacteria are often sensitive to a variety of environmental conditions, including chemical elicitors, pH, oxygen levels, temperature and light. Anaerobic bacteria can be effective in detecting certain types of tumors due to their anoxic conditions [288]. In a study in previous years, S. typhimurium cells were attached to liposomal particles. Studies have shown that bacteria can sense and swim towards cancer cells or cancer cell lysates [289]. For medical applications in the human body, where chemistry is everywhere. The study of Metin Sitti et al. proposes a model to simulate the 3D movement of bacteria-driven microswimmers, further elucidating the bacterial propulsion and chemotaxis of bacteria-driven microswimmers [290]. Martel et al. demonstrate the chemotactic migration behavior of the magnetotactic bacterium Marine *Magnetococcus* strain MC-1, which can be used to transport drug-laden nanoliposomes to hypoxic regions of tumors [291].

Above, we introduced the deformability of microrobots, which can not only be used to measure the mechanical properties of cells and create complex structures of tissues or organs but also to provide multimodal locomotion strategies and facilitate the transportation of goods. Ren et al. propose various control and performance enhancement strategies to let sheet-shaped soft millirobots achieve multimodal locomotion, including rolling, undulatory crawling, undulatory swimming and helical surface crawling, depending on different fluid-filled confined environments [292]. Zhiqiang Zheng and other teams present a shape-morphing strategy for microrobotic end-effectors madeto adapt to different physiochemical environments [293].

## 5. Summary

In the past few decades, microrobots have made tremendous progress and breakthroughs. From the emergence of conventional robots at the very beginning, gradual development has now witnessed robotics research evolve to miniaturized, intelligent and integrated microrobots. Although conventional robots are currently based on rigid drives such as motors and engines, they can show great efficiency and popularity in most macrosituations. But reducing these drives to millimeters or even smaller can lead to exponential manufacturing difficulty, significant performance degradation and inability to be used in body fluids or even some of the delicate manipulations of the human body. Therefore, biorobots based on living cell drives and even more versatile biohybrid robots benefit from the inherent microscale self-assembly of living tissues and high energy efficiency. This can effectively overcome these shortcomings and has the advantages of energy efficiency, reliability, controllability and biocompatibility at the microscale, and it can also minimize the potential toxic effects of biomaterials when applied to the human body. This makes them attractive for targeted therapeutics in vivo and is receiving increasing attention and research.

This paper summarizes recent advances in the use of biorobots composed of active single cells, such as algae, bacteria and cells, including their materials and actuation methods. Various novel biocompatible, biodegradable materials have been used in recent years to build microrobots that combine motile microorganisms with artificial components, and nature provides a rich database for science. Through natural selection, real-life organisms have evolved complex biological materials in which microalgae, bacteria and cells have different structures and rich functions, including hydrophobic/hydrophilic, biodegradable, magnetic, phototropic, lightweight and self-fluorescence, which have attracted widespread interest from researchers. Microrobots not only need the above characteristics but also the provision of power and controllability of motion for further realization of complex functions. To drive the movements of microrobots and then be able to control them to perform specific functions in the human body, we need to provide them with control information and the energy they need to drive. Micro/nanorobots are reviewed based on having exogenous or endogenous dynamics to drive facilitated directed motion. Some self-propelled micro-nanorobots composed of biological structures and synthetic materials (e.g., hydrogels, etc.) are usually supported by endogenous chemical reaction energy and have the advantages of low cost and ease of operation. However, the obvious disadvantage is that the direction of movement cannot be correctly controlled and proceeds with the reaction. In the later stage of driving control, there will be obvious insufficient operating power, resulting in difficult driving. At the same time, adaptive particles driven by other stimuli, such as chemistry, also have safety and biocompatibility problems of fuel and drive reaction products. In contrast, biohybrid microrobots driven by external stimuli, such as magnetic, light, acoustic and other external stimuli, typically enable microrobots to perform remote, dexterous, precise, fast and robust actuation and control. This stimulation can also be used to trigger the release of payloads when the microrobot reaches a specific location, improving the effective release of the treatment with the targeted drug. Meanwhile, the system is biocompatible and has significant advantages in terms of in vivo immune escape and enhanced targeting ability, while acoustic tweezers, employing the same wavelength as medical ultrasound, present a notably safer profile. Each of the three has its own advantages, disadvantages and complementarities. Their application in vivo and in vitro is described, where cell sorting is essential for cell observation, medical diagnostics, clinical applications and molecular biology research. Precise cell manipulation techniques have been developed for cell analysis at the single-cell level. This can provide small medical tools for biomedical applications such as disease surveillance, targeted drug delivery, minimally invasive surgery, etc. Manipulated by various drive systems, these robots exhibit different movement patterns for achieving various functions that facilitate the clinical applications of micro/nanorobots.

Although micro/nanorobots have great promise, the current use of single-cell manufacturing technology to form biohybrid robots has been greatly developed. Current research is mainly based on fluid environments and in vitro experiments, and in vivo research is still in an immature stage. There are various difficulties that limit its application in vivo; for example, current studies seem to attempt to track microrobots in real time to control them in vivo. For example, it is still difficult to track microrobots in real time to control their position and other functions in the body. The main challenges currently faced in the fabrication of biobased microrobots are the problems of finding suitable building materials, reliable energy supply and large-scale production. The materials used to make biobased microbots must be biocompatible, flexible and degradable. In this way, the biobased microrobots will not harm the organism or cause an immune response when they come into contact in vivo. In addition, these materials should be able to withstand the harsh conditions of the biological environment, such as temperature and pH fluctuations, among others. Biobased microrobots require a reliable power source, since conventional batteries are often too large or not biocompatible. Researchers need to explore alternative power sources, such as using biofuels or harnessing energy from the surrounding environment. But these methods are still in the early stages of development. Large-scale manufacturing of biobased microrobots is another challenge. Current manufacturing techniques tend to be time-consuming and more expensive. These factors limit the practical application of biobased microrobots. Biobased microrobot control is another challenge. The main challenge is the difficulty of achieving precise motion control in such a small area. And it is difficult to achieve precise sensing in complex biological environments. Due to the small size and limited resources of the designed robot, designing actuators that can transmit control, be compatible with the biological environment and be able to perceive the complexity of the surrounding environment is a key issue. Therefore, in summary, research on small-scale robotics is at a beginning stage and is also at a high level of innovation. While micro/nanorobots have the potential to perform complex tasks and many initial advances have been made, there are still many challenges and unresolved issues that prevent us from manipulating micro/nanorobots with multiple functions to perform a variety of tasks.

Bio-based microrobots have unique advantages over synthetic microrobots. Because of their simple access to raw materials and low price, they have attracted widespread interest among researchers. For microalgae microrobots, algae containing chlorophyll have autofluorescence properties, which is convenient for later observation. Most algae are very degradable and biocompatible and are suitable for human treatment. Different algae have different characteristics. Algae such as *Chlorella*, *Chlamydomonas reinthine* and *Spirulina* can also be used as drug carriers and can also be used for treatment in their own right, releasing reactive oxygen species that can help in the treatment of cancer. For bacteria, because of their flagella, they have a higher speed of movement and can get to the desired place more quickly. For cells, it can reduce the elimination of microrobots by the human immune system and improve the therapeutic effect. Microrobots have unique advantages in targeted delivery and minimally invasive surgery, and the treatment of cancer is usually chemotherapy and radiotherapy, and the development of microrobots may make cancer treatment simpler. Although there is still a long way to go to develop powerful microrobots from theory to the operating table, the development of related research in recent years has brought fantasy closer to reality. Despite the many challenges to overcome, current biorobotics technology has also made exciting progress, laying the foundation for further research and applications in the future.

## Figures and Tables

**Figure 1 micromachines-14-01710-f001:**
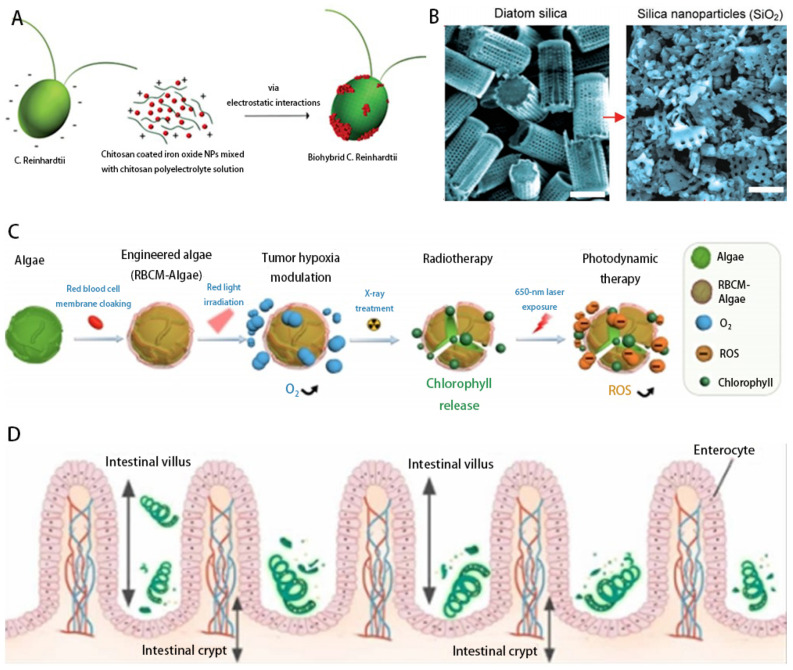
Classification of microalgal microrobots: (**A**) Biohybrid algal microrobots using *Chlamydomonas reinhardtii* as a carrier [49]. (**B**) Diatoms as a system for drug delivery package [24]. (**C**) Innovative approach to overcome tumor hypoxia using the natural photosynthetic system of *Chlorella* [50]. (**D**) Use of *Spirulina* as a drug carrier [25].

**Figure 2 micromachines-14-01710-f002:**
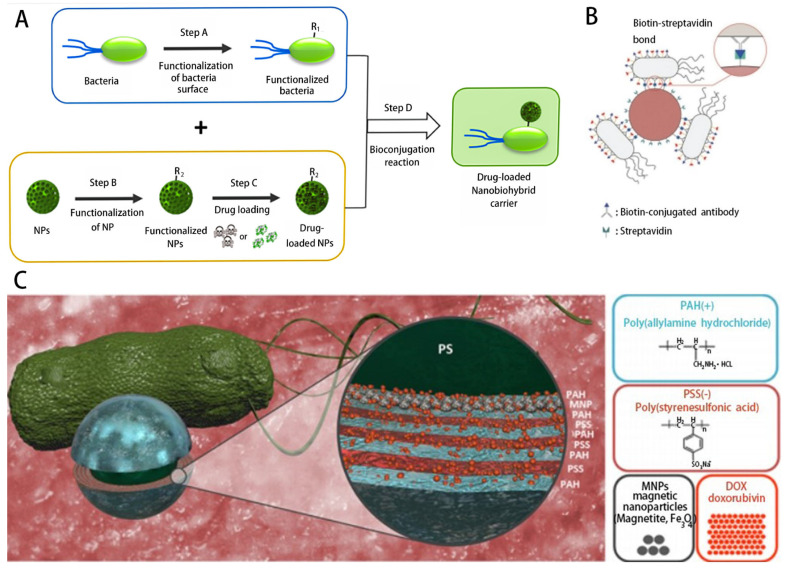
Different types of bacterial microrobots: (**A**) Bacteria-based microrobot [51]. (**B**) *E. coli* bacteria type-I-based microrobot [67]. (**C**) Multifunctional bacteria-driven microrobot [52].

**Figure 4 micromachines-14-01710-f004:**
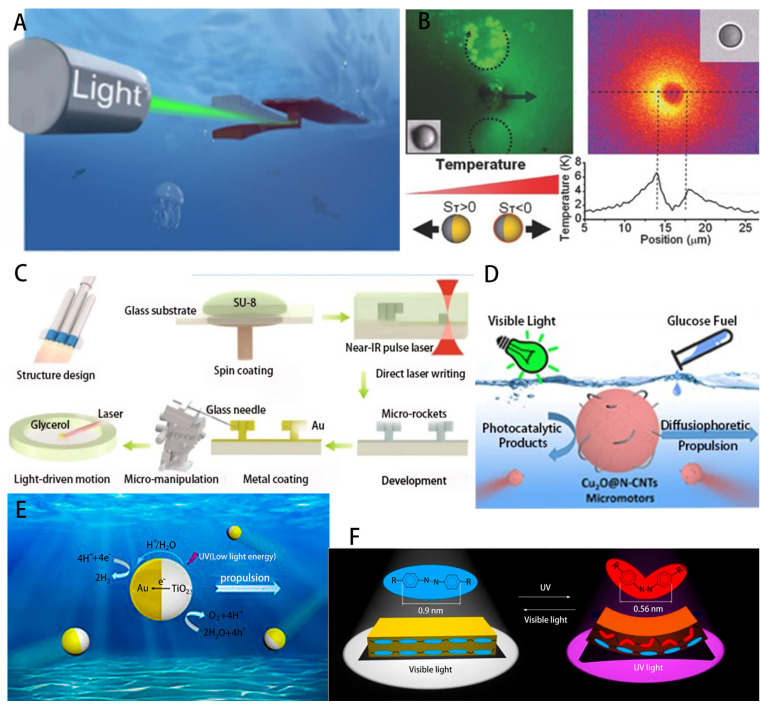
Micro/nano motor system based on light-driven swimming: (**A**) Self-excited oscillation of a soft swimming robot (OsciBot) powered and controlled by visible light [146]. (**B**) Self-propulsion of microscale Janus particles in semi-metallic-coated colloids under laser irradiation [69]. (**C**) Design and fabrication of a microrocket robot with all-optical drive and blood tracking by 3D printing [145]. (**D**) An efficient copperoxide@N-doped carbon nanotube (Cu_2_O@N-CNT) micromotor fueled by glucose for environmentally friendly visible-light photocatalysis [152]. (**E**) An efficient light-driven T with wireless steering and speed control [153]. (**F**) Liquid crystal thin films (LDLCF) containing azobenzene chromophores with cis–trans isomers can be bent by ultraviolet (UV) and recycled by visible light, serving as “motors” for small swimming soft robots that can perform complex movements [148].

**Figure 5 micromachines-14-01710-f005:**
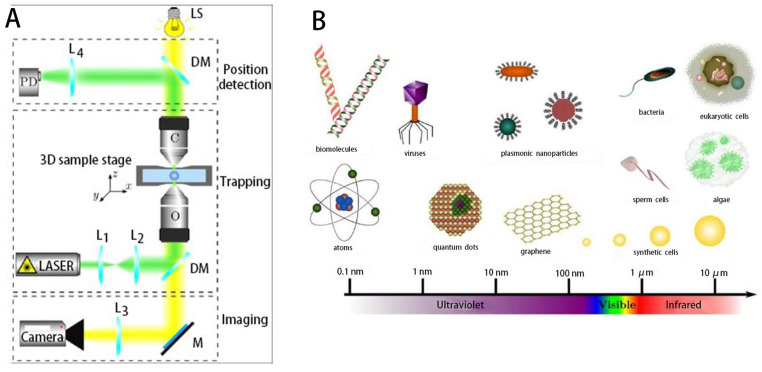
Manipulation of a wide range of particles using optical tweezers: (**A**) The basic optical tweezer setup consists of three parts: capture optics, imaging optics, and position detection optics [174]. (**B**) A wide range of objects can be captured using optical tweezers, from individual atoms and molecules to particles and microorganisms [174].

**Figure 6 micromachines-14-01710-f006:**
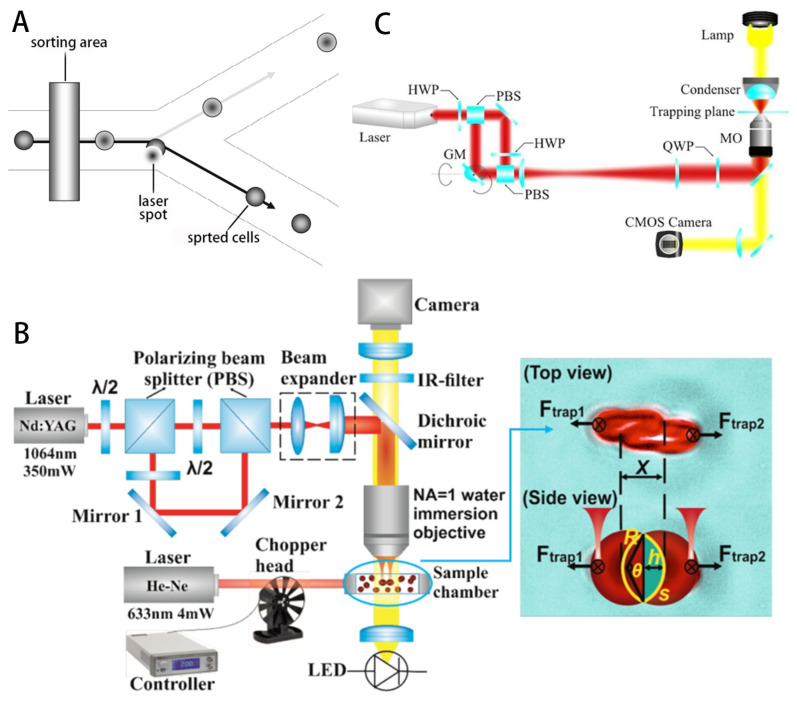
Optical-tweezer-based particle manipulation: (**A**) Improved microfluidic channel structure (intersection region) for rapid cell sorting [137]. (**B**) Schematic layout of the double-channel optical tweezers system combined with a chopper-modulated pulsed laser irradiation module for studying RBC interaction and laser irradiation effects [193]. (**C**) Study of red blood cell (RBC) deformability in the presence or absence of diabetic retinopathy (DR) with a dual-beam schematic diagram of the optical tweezer experimental setup [194].

**Figure 7 micromachines-14-01710-f007:**
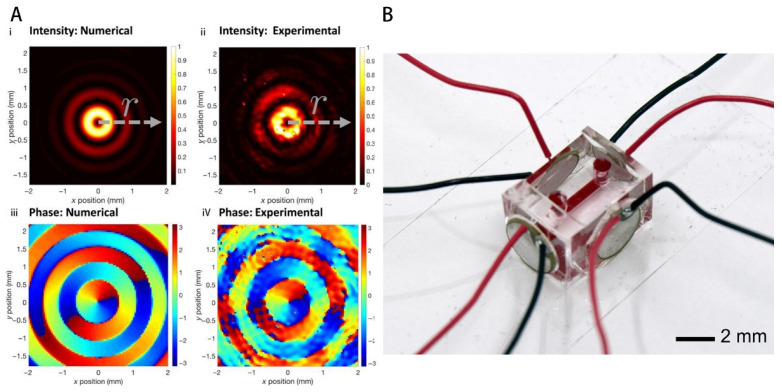
Examples of traveling-wave and standing-wave acoustic tweezers: (**A**) Comparison of simulated and actual effects of acoustic potential traps [126]. (**B**) Micrograph of the in vivo acoustic manipulation chamber [113].

**Figure 8 micromachines-14-01710-f008:**
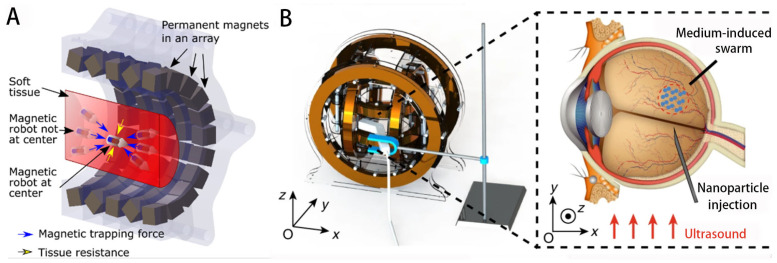
Permanent magnet and electromagnetic drive systems: (**A**) The permanent magnet drive system [101]. (**B**) The Helmholtz solenoid coil device generates a rotating magnetic field [105].

**Figure 9 micromachines-14-01710-f009:**
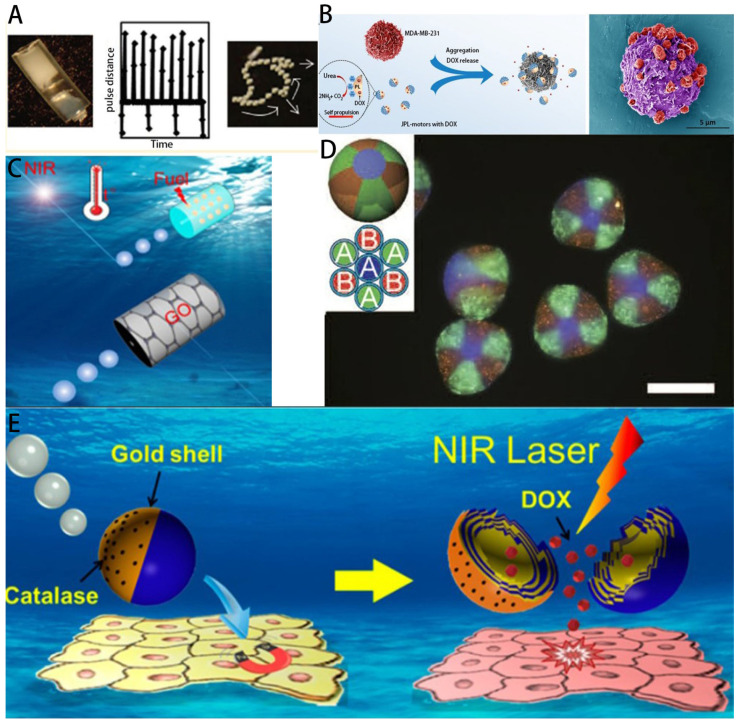
Examples of microrobots based on other types of stimulus drive: (**A**) Self-propelled particles made of ethanol-infused polyacrylamide hydrogels contained in plastic tubes moving in an oscillatory mode by the Marangoni effect [237]. (**B**) An endogenous enzyme-driven Janus platelet micromotor (JPL-motor) on MDA-MB-231 breast carcinoma cells after chemical drug loading schematic diagram of targeted delivery [224]. (**C**) Preparation of an emulsion hydrogel motor (E-H motor) consisting of a low-boiling oil fuel and a hydrogel matrix by an oil-in-water (O/W) emulsion templating method that can be efficiently propelled by thermally stimulated bubble generation [221]. (**D**) A three-dimensional multichamber (3D-MC) particle with complex shapes generated by deformation control using induced diffusion flow and Marangoni flow [228]. (**E**) Schematic of an enzymatic Janus micromotor. The micromotor can be loaded with Adriamycin and triggers drug release by an external near-infrared laser (NIR) whose motion depends on the reaction of Pt with hydrogen peroxide [234].

**Figure 10 micromachines-14-01710-f010:**
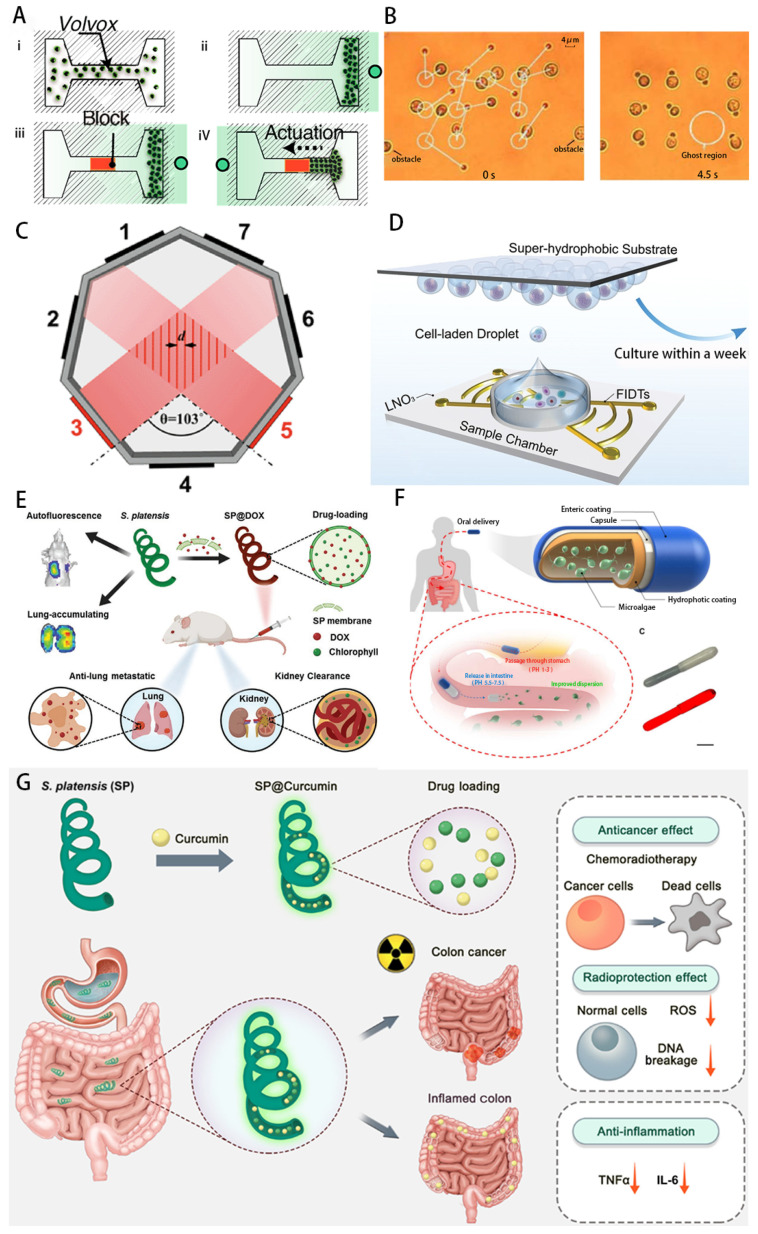
Single-cell microrobotics in cells, organs and tissues: (**A**) Experimental procedure of driving a movable block by microalgae using their own phototropism [144]. (**B**) Snapshots of moving two groups of microparticles into arrays with an artificial potential field-based controller [245]. (**C**) Using acoustic tweezers to arrange cells [268]. (**D**) Using acoustic droplet printing device to build tissue construction [276]. (**E**) Biohybrid system for DOX-loaded *Spirulina* [281]. (**F**) Algal-motor-loaded capsule system [48]. (**G**) Microalgae-based oral drug delivery system [25].

**Table 2 micromachines-14-01710-t002:** Outfield drive methods and characteristics.

Power	Penetration Capability	Advantages	Disadvantages	Classification	Features	Reference
Magnetic fields	Strong	Strong control and spatial resolution, high robustness	Metals added to the core material or surface coating will limit its application in vivo	Permanent magnet system	Low energy loss, greater convenience and stronger magnetic field	[101,102,103]
				Electromagne-tic drive system	Free to change direction and intensity	[81,104,105,106,107,108,109]
Soun fields	penetrate the entire human body	Capable of contactless control, low power, small size and high safety	The application of ultrasound might cause oxidative stress in cells	Standing wave	High precision, simple system and low cost	[110,111,112,113,114,115,116,117,118]
				Traveling wave	High flexibility and large operating space	[119,120,121,122,123,124,125,126,127]
				Acoustic flow	Ability to manipulate individual objects in groups	[128,129,130]
Light fields	penetrate up to 1–2 cm under the skin	High instantaneous drive energy, reflecting a certain degree of programmability; noninvasive	The stability and controllability of light control are not strong; cannot penetrate tissues	Phototropism	Organisms have a natural phototropism	[46,131,132,133]
				Photothermal or photocatalytic	Driving nanorobots by light-energy-induced catalytic reactions inside the robot	[69,134,135]
				Optical tweezers	Manipulating cells or organisms at the microscopic scale with high flexibility	[69,136,137,138,139,140,141,142,143]

## Data Availability

Not applicable.

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
