# Peer review of "A Review of Single-Cell Microrobots: Classification, Driving Methods and Applications"

_micromachines, 2023, doi:10.3390/mi14091710_

Round 1

Reviewer 1 Report

This paper proposed a comprehensive review on single-cell microrobots from three aspects: biological and design processing often employed in biotype micro- and nano-robots, principles and effects of external field actuation, and applications of micro- and nano-robots in vivo and in vitro. However, the purported significance of the paper on reviewing single-cell microrobots is not explicitly stated, and the discussion of challenges and future significance is not insightful enough. There are some queries should be explained and relevant parts need to be amended:

1.       Sentences in many important sections of this manuscript are too long to cause grammatical errors and logical fallacies. For example, sentence ‘The more novel…human body is minimal’ contains over 70 words in the second paragraph of Introduction. Sentence ‘Although micro/nanorobots…functions in vivo’ contains over 90 words in the last paragraph of Summary. The authors should sort out the logical relationship between the text and polish the English. The presentation of this article in terms of logical expression and English writing needs to be greatly improved.

2.       Exhaustive comparison between microrobots with and without biological components is required in section Introduction to illustrate why ’the more novel microbial-based micro-nano-robots are integrated micro-robots composed of biological and artificial components’. It is suggested that the authors supplement their research in this aspect and consult and reference relevant literature on this topic. The following literatures can be referred to:

1)       Y Sun, et al. Robotic micromanipulation: Fundamentals and applications[J]. Annual Review of Control, Robotics, and Autonomous Systems, 2019, 2: 181-203.

2)       Y Hou, et al. A review on microrobots driven by optical and magnetic fields[J]. Lab on a Chip, 2023.

3)       M Sitti, et al. Biohybrid actuators for robotics: A review of devices actuated by living cells[J]. Science robotics, 2017, 2(12): eaaq0495.

4)       L Liu, et al. Development of micro-and nanorobotics: A review[J]. Science China Technological Sciences, 2019, 62: 1-20.

3.       Disadvantages and application limitations of diverse types of microrobot and outfield drive methods are not illustrated in Table 1 and Table 2.

4.       It is a bit far-fetched for constructing tissues or organs and drug delivery to be collectively classified as organ and tissue analysis in Section 4.2. It is recommended that these two parts be written separately.

5.       Too many grammatical errors in the manuscript, like ‘In addition, enables easy’ in the third paragraph of section Introduction.

6.       The deformability of microrobots can be used not only to measure the mechanical properties of cells and to fabricate complex structures of tissues or organs, but also to provide multimodal locomotion strategies and to facilitate cargo delivery. It is suggested that the authors supplement their research in these aspects and consult and reference relevant literature on this topic. For example:

1)       Xu T., et al. Multimodal locomotion and cargo transportation of magnetically actuated quadruped soft microrobots[J]. Cyborg and Bionic Systems, 2022, 2022: 0004.

2)       Zheng Z, et al. Programmable aniso-electrodeposited modular hydrogel microrobots[J]. Science Advances, 2022, 8(50): eade6135.

3)       Sitti M., et al. Soft-bodied adaptive multimodal locomotion strategies in fluid-filled confined spaces[J]. Science advances, 2021, 7(27): eabh2022.

4)       Wang H, et al. Ionic shape-morphing microrobotic end-effectors for environmentally adaptive targeting, releasing, and sampling[J]. Nature communications, 2021, 12(1): 411.

5)       Nelson B. J., et al. Magnetic continuum device with variable stiffness for minimally invasive surgery[J]. Advanced Intelligent Systems, 2020, 2(6): 1900086.

7.       In addition to droplet-based bioprinting, micro-extrusion, and laser-assisted printing, photolithography is also an important method to efficiently construct tissues or organs. The following literatures can be referred to:

1)       You S, et al. High cell density and high-resolution 3D bioprinting for fabricating vascularized tissues[J]. Science Advances, 2023, 9(8): eade7923.

2)       Park C., et al. Precisely printable and biocompatible silk fibroin bioink for digital light processing 3D printing[J]. Nature communications, 2018, 9(1): 1620.

3)       Li X, et al. Holographic display-based control for high-accuracy photolithography of cellular micro-scaffold with heterogeneous architecture[J]. IEEE/ASME Transactions on Mechatronics, 2021, 27(2): 1117-1127.

4)       Kim S. H., et al. 3D bioprinted silk fibroin hydrogels for tissue engineering[J]. Nature protocols, 2021, 16(12): 5484-5532.

8.       In the summary section, the authors only mentioned the challenge that microrobots are difficult to track in vivo, which is a common issue for all in vivo microrobots. It is suggested to provide an exhaustive discussion on challenges and issues specific to manufacturing and controlling of bio-based microrobot.

9.       It would be better if the authors can provide some inspiring discussion on future work of bio-based microrobot such as research directions with great potential and value, crucial technical issues to be resolved.

Moderate editing of English language is required.There are many very long sentences with grammatical errors in this manuscript.

Author Response

Please refer to the annex

Reviewer 2 Report

The review authored by Yuhang et al. provides a succinct overview of single-cell microrobots, which are minute devices combining individual cells with artificial components. The article extensively examines diverse types, control techniques, and applications of these microscale robots within the realm of biomedicine. Noteworthy instances of such single-cell microrobots encompass algae, bacteria, and cells proficiently propelled by external stimuli like light, sound, and magnetic fields. In addition, the review critically assesses the merits and obstacles associated with single-cell microrobots concerning their utility in precise drug delivery, minimally invasive therapeutic interventions, and other assorted tasks. The primary objective of this article is to furnish a holistic portrayal of the present status and potential trajectories in the field of single-cell microrobotics. The manuscript is well-written and well-organized. I only have one suggestions about this review.

The authors may provide a quantitative comparison of the performance and efficiency of different biohybrid micro-robots, such as their speed, power, durability, and biocompatibility. 

Minor editing of English language required

Author Response

Please refer to the annex

Reviewer 3 Report

This paper delivers an exhaustive review on single cell microrobots, which are intricate devices resulting from the integration of individual cells with artificial components. It systematically categorizes these microrobots into groups such as cell-based, bacteria-based, and algae-based, providing insights into their design, fabrication, and the pivotal role of external field-driven technologies — including optical, acoustic, and magnetic — in their operation. Additionally, the paper delves into the real-world applications of these robots in areas like precision delivery and minimally invasive therapy. It culminates with a succinct summary, shedding light on present challenges and mapping out future trajectories in microbial-based robotics.

In essence, this review offers a meticulous and in-depth exploration of the realm of single cell microrobots. The authors brilliantly elucidate the varied classifications and crafting techniques of these microrobots while highlighting the significance of external field-driven technologies. Teeming with valuable insights, this piece stands as an indispensable guide for both budding and seasoned researchers in the field. But to further enhance this paper, I still have some suggestions as following:

1.      In the introduction, the authors state, “Moreover, external field drives have a high spatial resolution, and recent research on driving in three dimensions has overcome the limitations of one and two dimensions, offering micro and nano robots enhanced flexibility and functionality.”

Statements within the introduction should be substantiated with evidence. Therefore, I advise the authors to incorporate pertinent references at this juncture.

2.      Again, in the introduction, between lines 64 to 67:

Original: "In addition, enables easy capture," Suggested: "In addition, it facilitates easy capture,"

Original: "rotation, and stretching of cells using the micro-nano robotic system," Suggested: "rotation, and stretching of cells utilizing the micro-nano robotic system,"

3.      The third section of this paper is labeled “Outfield drive” on line 288. I believe the intended term is “external field drive”.

4.      Within the “3.3 Magnetic drive” subsection, the authors enumerate various magnetic drive methodologies, which include: “custom-designed or Helmholtz coils, magnetic resonance imaging (MRI) gradient coils within MRI scanners, coil arrays, or permanent magnet (e.g., single or dual dipole permanent magnets, Halbach arrays) systems [187].”

However, in “3.3.2 Electromagnetic drive system”, there seems to be a predominant focus on Helmholtz magnetic coils. To my knowledge, numerous magnetic systems don't rely on Helmholtz coils.

A comprehensive discussion encompassing diverse magnetic systems, delineating their theoretical foundations, generated magnetic fields, and their respective pros and cons would be beneficial. Focusing solely on electromagnetic drive systems with Helmholtz coil arrays lacks depth in covering the entirety of existing electromagnetic drive systems.

Here are my refined recommendations for various magnetic manipulation systems for your reference:

                               I.            Rotating magnetic field:

1)      Z. Ye, S. Regnier, and M. Sitti, “Rotating magnetic miniature swimming robots with multiple flexible flagella,” IEEE Trans. Robot., vol. 30, no. 1, pp. 3–13, Feb. 2014.

2)      M. Salehizadeh, Z. Li and E. Diller, "Independent Position Control of Two Magnetic Microrobots via Rotating Magnetic Field in Two Dimensions," 2019 International Conference on Manipulation, Automation and Robotics at Small Scales (MARSS), pp. 1-6, doi: 10.1109/MARSS.2019.8860954, 2019.

3)      J. Park, J.Y. Kim, S. Pane, B. J. Nelson, and H. Choi, H,” Acoustically mediated controlled drug release and targeted therapy with degradable 3D porous magnetic microrobots.” Advanced healthcare materials, 10(2), 2001096, 2021.

                            II.            Gradient magnetic field:

1)      J. Giltinan, and M. Sitti, “Simultaneous six-degree-of-freedom control of a single-body magnetic microrobot”, IEEE Robotics and Automation Letters, 4(2), pp.508-514, 2019.

2)      D. Li, F. Niu, J. Li, X. Li and, D. Sun, “An Electromagnetic Actuation System with New Core Shape Design for Microrobot Manipulation,” IEEE Transactions on Industrial Electronics, vol. 67, pp. 4700-4710, 2020.

3)      3.   Z. Yang, and L. Zhang, “Magnetic actuation systems for miniature robots: A review”, Advanced Intelligent Systems, 2(9), p.2000082, 2020.

                         III.            Alternating magnetic field:

1)      H. Xie, M. Sun, X. Fan, Z. Lin, W. Chen, L. Wang, L. Dong, and Q. He, “Reconfigurable magnetic microrobot swarm: Multimode transformation, locomotion, and manipulation”, Science robotics, 4(28), p.eaav8006, 2019.

2)      L. Xing, D. Li, H. Cao, L. Fan, Z. Liu, Z. Li, and D. Sun, “A New Drive System for Microagent Controls in Targeted therapy,” Advanced Intelligent Systems, 4(9), 2100214, 2022.

5.      Furthermore, in “3.3.2 Electromagnetic drive system”, from lines 666 to 672, the authors observe, “The electromagnetic system primarily employs an electromagnetic coil to generate a magnetic field”.

Typically, electromagnetic manipulation systems utilize coil arrays for magnetic field generation. Adjusting the current to each coil enables these systems to produce varied magnetic fields or modify the magnetic flux and gradient. To ensure clarity for readers, I recommend a precise definition of the system.

6.      From line 975 to line 976, the statement reads: "While acoustic tweezers use the same wavelength as the medical application of ultrasound, which boasts a superior safety profile."

For clarity, I suggest: "Acoustic tweezers, employing the same wavelength as medical ultrasound, present a notably safer profile."

7.       In the summary part, the content spanning lines 985 to 991 could benefit from restructuring for enhanced coherence.

Listed in the comments and suggestions for authors.

Author Response

Please refer to the annex

Round 2

Reviewer 1 Report

The authors have improved the manuscript by following the reviewers' comments. I suggest for publication.